# Temporal social network modeling of mobile connectivity data with graph neural networks

**Joel Jaskari**[1], **Chandreyee Roy**[1], **Fumiko Ogushi**[2], **Mikko Saukkoriipi**[1], **Jaakko Sahlsten and**[1], **Kimmo Kaski**[1]*

1 Department of Computer Science, Aalto University, Aalto, Finland, 2 Faculty of Mathematical Informatics, Meiji Gakuin University, Kamikurata-cho, Totsuka-ku, Yokohama, Kanagawa, Japan

* kimmo.kaski@aalto.fi

## Abstract

Graph neural networks (GNNs) have emerged as a state-of-the-art data-driven tool for modeling connectivity data of graph-structured complex networks and integrating information of their nodes and edges in space and time. However, as of yet, the analysis of social networks using the time series of people's mobile connectivity data has not been extensively investigated. In the present study, we investigate four recently proposed snapshot - based temporal GNNs in predicting the phone call and SMS activity between users of a mobile communication network. In addition, we develop a simple non - GNN baseline model using recently proposed EdgeBank method. Our analysis shows that the ROLAND temporal GNN outperforms the baseline model in most cases, whereas the other three GNNs perform on average worse than the baseline. The results show that GNN based approaches hold promise in the analysis of temporal social networks through mobile connectivity data. However, due to the relatively small performance margin between ROLAND and the baseline model, further research is required on specialized GNN architectures for temporal social network analysis.

## Introduction

Humans as social beings belong to various offline or online social networks, which have emerged as a result of mutual relationships based on communication [1,2], cooperation [3], or other types of social interaction between them [4–7]. The structure of these social networks has a tendency to evolve or change in the course of the lives of people [8–11], as their social ties with others are dependent on several factors such as age and gender [12–14] along with socio-economic status and living area, among others [15,16]. Since these social ties are based on multiple aspects of relationships, social networks are multilayered [17,18] and have evolving community structures. In the literature, social networks are generally modeled as graph - like structures with individuals, groups, or organizations as nodes, and the relationships or ties between them as links or edges [19]. Furthermore, both the nodes and edges

**Data availability statement:** The dataset generated and/or analyzed during the current study are not publicly available due to a signed NDA, because it could potentially be used to identify anonymized individuals. This dataset is available upon reasonable request from data contact and data co-owner Professor Mikko Kivelä, mikko.kivela@aalto.fi.

**Funding:** The author(s) received no specific funding for this work.

**Competing interests:** The authors have declared that no competing interests exist.

may be dynamic or temporal in nature. Modeling human relationships can enable us to understand and even predict how people interact with each other [20], and to build strategies that may lead to improvements in different aspects of life [8]. Such modeling approaches include, for example, the forecasting of epidemics under various mitigation strategies [21,22], modeling the perceptions of vaccinations [23], and urban planning [5].

Mobile phone data sets have been used to get insight into the structure and dynamics of social networks from the level of individuals to groups, communities, and societies up to the of level of populations [24]. These population-level datasets contain information about individuals' mobile service usages in terms of the amount, frequency, and duration of calls and text messages (SMS) sent and received, as well as location based on cell tower position. These datasets are powerful in capturing and predicting service users' relationships through their past behaviour, which can be used to quantify the closeness of relationships, detect social ties and groups [25], as well as the reciprocal nature of caller-callee pairs [26].

Human communication is generally bursty in nature [27] and several differences are observed with respect to the age and gender of the caller-callee pairs. In fact, younger individuals have been observed to have larger social networks and higher churn rates compared to the older population [12,13]. Compared to women, men have been found to have larger networks of friends, while being less talkative on the phone [28]. In addition, some studies have observed distinct circadian rhythm patterns through mobile phone datasets [29–31]. Furthermore, there is an added advantage of having location-based mobile phone data as it allows researchers to study the patterns of mobility of individuals [15,32].

Recently, Graph Neural Networks (GNNs) have emerged as a state-of-the-art deep learning approach for analysing connectivity data of graph-structured complex networks. These models facilitate the classification and regression of attributes of nodes, edges, and entire graphs [33]. A special case of a GNN prediction task is link prediction, i.e., the prediction of the existence of an edge between pairs of nodes. Temporal, or dynamic, GNNs have been developed for graphs that are temporal in nature, such that the nodes and edges, or their attributes, evolve over time [34].

In the literature, GNNs have been utilised for applications ranging from computer vision to physics [35] and weather forecasting [36]. They have also been proposed for various tasks in social network analysis, e.g., to forecast social influence [37], social recommendation [38], criminal network analysis [39], and fake news detection [40]. In the context of mobile phone datasets, temporal GNNs have been applied to cellular traffic prediction [41]. However, the case of predicting multi-dimensional edge attributes of temporal social networks, such as the amount of future bilateral calls and text messages, is to the best of our knowledge yet unstudied. Such GNNs could be directly used in e.g., network traffic prediction and social recommendation, but also probed with explainable artificial intelligence methods to gain insight into the patterns that influence temporal social networks.

In the present study, we consider temporal graph neural networks for modeling monthly mobile phone activity between the users of a mobile phone network. We utilise a unique mobile phone dataset from a European country that spans over a

time period of three years, collected between 2007–2009. The contributions of this work are as follows. First, we present temporal graph characteristics of our temporal mobile phone dataset to yield insight to the dynamics of the network edges. Second, we propose a modification to the recently proposed EdgeBank temporal edge prediction baseline [42], which we call *rEdgeBank*, to make it suitable to serve as a baseline in the prediction of multi-valued edge attributes. Third, we make a comprehensive evaluation of the prediction accuracy of four state-of-the-art temporal graph neural networks that have been proposed in the literature, namely the Graph Convolutional Recurrent Networks [43], Variational Graph Recurrent Neural Networks [44], Dynamic Self-Attention Networks [45], and ROLAND [46], for the prediction of the quantity of future calls and text messages between pairs of individuals. Specifically, we systematically evaluate the prediction accuracy of the number of calls and text messages under three different paradigms suggested in literature. In addition, we stratify the results based on the age and gender for a more fine-grained analysis. A graphical illustration of our approach is presented in Fig 1.

## Materials and methods

In this section, we first describe our mobile phone dataset in detail. We then describe graph neural networks and their temporal variants to introduce background on the subject. Lastly, we go over the four temporal graph neural network methods and evaluation measures used for our results.

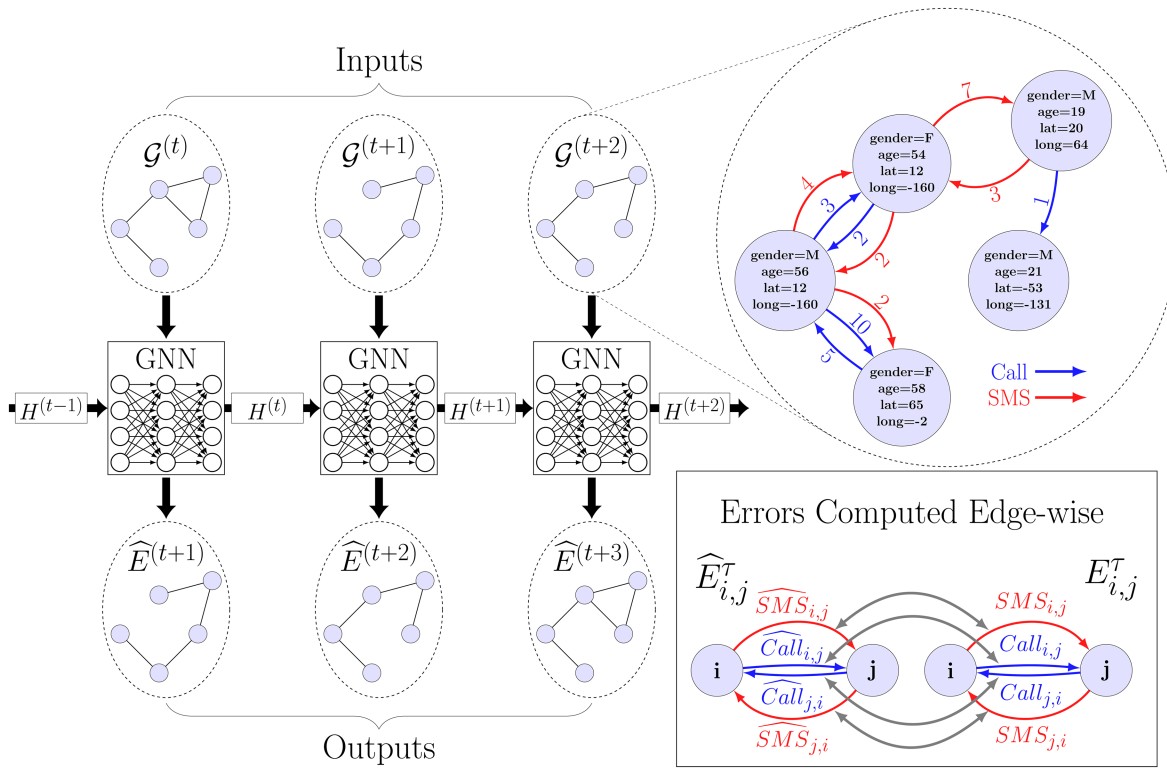

**Fig 1**. **Graphical illustration of the approach. Temporal graph neural network (GNN) processes each temporal graph snapshot $\mathcal{G}^{(\tau)}$ independently, but also leverages information about the past state of the temporal graph through the recurrent connection $H^{(\tau)}$ that it learns to produce.** Each snapshot of the temporal graph consist of the node features (i.e., the gender, age, latitude, and longitude) and the number of bidirectional calls and text messages between the nodes. The GNN output for input snapshot $\mathcal{G}^{(\tau)}$ and past hidden state $H^{(\tau-1)}$ is a prediction of the call and SMS activity for the next month $\tilde{E}^{(\tau+1)}$.

## Data

The mobile phone dataset consists of anonymized individual call detail records (CDRs) of phone calls and SMS made between 2007 to 2009 in a European country. The CDRs comprise the ego-alter pair of individuals who made a voice call or sent a SMS, along with information about the date, time, and the direction of the mobile activity. In addition to the mobile activity data, the CDRs had information about the individuals regarding their age, gender, and the most common location of calls, obtained during the year 2007, in the form of latitude and longitude.

We performed multiple preprocessing steps to standardise the dataset for our analyses. Due to the low number of young and old individuals, we only considered people aged between 18 and 65. Individuals with missing information regarding the age, gender, latitude, or longitude were also filtered out. To avoid users potentially associated with call centers or subscription sellers, we excluded the users with a daily average call activity exceeding 8 calls. The mobile company provider had anonymised the users with random unique identifiers before handing the data over to us to study under the NDA. We observed that some of the users were only active for a single year, which we suspected to have been caused by some identifiers being randomly renamed by the provider on the start of a new year. Thus, we filtered out the users by only considering those that had at least one event of activity in every year. The reasoning for this filtering was that the data-driven graph neural networks cannot distinguish naturally occurring end of mobile activity between people from a user being assigned to a new random identifier. Afterwards, each ego-alter pair's total number of calls and text messages were aggregated monthly, so the graph evolves over a time span where the month, or timestep, $t$ runs from 1 to 36.

Finally, the egocentric graphs were combined to a single large temporal graph. The nodes of the graph were associated with a vector of four features that included the age, gender, latitude, and longitude. The temporal edges were formed as follows. For each month $t$, the observed ego-alter pairs were determined to have a directed edge between them. To allow the graph neural networks to utilise both sent and received mobile activity information for every node, we made four features for each directed edge: the number of calls and text messages from ego to alter and from alter to ego. In the end, there were thus 36 graphs that represented the state of the mobile phone activity network for each month. It is worth noting that as the edges were made on a month by month basis, it is possible that an edge either appears, disappears, or remains from month to month. The resulting dataset had 247 732 nodes with 5 454 077 temporal edges including 80 665 293 calls and 16 372 610 text messages.

The temporal graph was divided to training, validation, and test subsets, such that the data from January 1, 2007 to December 31, 2008 belonged to the training set, data from January 1, 2009 to June 30, 2009 to the validation set, and data from July 1, 2009 to December 31, 2009 to the test set. Due to the extremely large size of the graph, we preprocessed the graph by sampling a "3-hop" subgraph for each of the 247 732 nodes, i.e., selecting each node as a seed and expanding the graph from that seed node for all the nodes and edges that have a path length of up to three.

## Temporal graph neural networks

In this subsection, we will introduce the background on graph neural networks and their temporal variants. We will then describe in detail the models we use for our experiments. Lastly, we shall describe the evaluation measures used to assess the performance of the models.

**Graph neural networks.** Graph neural networks are deep learning models that operate on data represented as graphs [33], i.e., data that consists of nodes and edges, and possibly their associated features. We shall denote a graph as $\mathcal{G} = (\mathcal{X}, \mathcal{E}, X, E)$, where $\mathcal{X} = \{1, \ldots, n\}$ is the set of $n$ nodes, $\mathcal{E} \subseteq \mathcal{X} \times \mathcal{X}$ is the set of edges between the nodes, $X = \{x_i \mid i \in \mathcal{X}\}$ is the set of $d_x$-dimensional node features for the $n$ nodes, i.e., $x_i$ is the vector of features for the node $i$, and $E = \{e_{i,j} \mid (i,j) \in \mathcal{E}\}$ is the set of $|\mathcal{E}|$ edge features with dimensionality $d_e$. A temporal graph that consists of snapshot -type data is a set of graphs $\{\mathcal{G}^{(1)}, \ldots, \mathcal{G}^{(T)}\}$, where $\mathcal{G}^{(t)} = (\mathcal{X}^{(t)}, \mathcal{E}^{(t)}, X^{(t)}, E^{(t)})$, which describes the state of the temporal graph sampled at time points $t \in \{1, \ldots, T\}$, i.e., $\mathcal{G}^{(t)}$ is a snapshot of the temporal graph at time $t$.

Graph convolutional neural networks are GNNs that leverage convolution -like operations reminiscent of deep learning convolutional neural networks for images [35]. A graph convolution processes the graph and outputs new representations for each node:

$$H = G(\mathcal{G}, \theta),\tag{1}$$

where $H = \{h_i \mid i \in \mathcal{X}\}$, $h_i \in \mathbb{R}^{d_h}$, $\theta$ are the convolutional layer parameters, and $G(\cdot, \cdot)$ denotes the convolution operation. The two types of graph convolutions most relevant to this work are the message-passing and aggregation, and Chebyshev convolutional layers.

A message-passing and aggregation layer computes the following [46,47]:

$$\begin{aligned} H &= G_{mpa}(\mathcal{G}, \theta) \\ &= \{f_{agg}(\{f_{msg}(x_s, x_d, e_{s,d}, \theta) \mid s \in \mathcal{N}(d, \mathcal{E})\}) \mid d \in \mathcal{X}\}, \end{aligned}\tag{2}$$

where $s$, $d$ denote the "source" and "destination" nodes, respectively, $\mathcal{N}(d, \mathcal{E})$ is an operator that returns the neighbors of the node $d$ based on the set of edges $\mathcal{E}$, and $f_{msg}$ and $f_{agg}$ denote the so-called message and aggregation functions, respectively. The message function can be implemented with e.g., a neural network parameterized by $\theta$, whereas the aggregation function is chosen as a permutation invariant operation, such as maximum or average. Many implementations will also include the destination node representations within $G_{mpa}$, in order for the new representation to also be a function of the node itself in addition to its neighbors. This can be achieved by considering self-connections for each node.

The Chebyshev convolutional layers are based on the spectral domain convolution of a graph, where the Chebyshev polynomials, up to a chosen degree $K$, are evaluated at the scaled graph Laplacian and modulated with learned parameters [48]. Precisely:

$$\begin{aligned} H &= G_{cheb}(\mathcal{G}, \theta) \\ &= \sum_{k=0}^{K-1} \theta_k T_k(\tilde{L})X, \end{aligned}$$

where $\tilde{L}$ is the scaled and normalized Laplacian and $T_k(\cdot)$ is the Chebyshev polynomial of order $k$. The degree $K$ determines the localization of a Chebyshev convolutional layer such that the output for a node is dependent on all the input nodes with a path length of up to $K$, i.e., the layer has a receptive field size of $K$.

The graph convolutions can be iteratively applied to form a deep convolutional architecture for a GNN. Specifically, the node features that the first layer outputs are fed to the second layer, and so on:

$$\begin{aligned} \mathcal{H}_{\mathcal{G}1} &= (\mathcal{X}, \mathcal{E}, G(\mathcal{G}, \theta_1), E), \\ \mathcal{H}_{\mathcal{G}2} &= (\mathcal{X}, \mathcal{E}, G(\mathcal{H}_{\mathcal{G}1}, \theta_2), E), \\ &\vdots \\ \mathcal{H}_{\mathcal{G}l} &= (\mathcal{X}, \mathcal{E}, G(\mathcal{H}_{\mathcal{G}l-1}, \theta_l), E), \end{aligned}$$

where each graph convolutional layer is associated with its own learnable parameters. A message-passing and aggregation layer will only propagate information between neighbors of nodes, and thus the receptive field can be increased by multiple layers, where e.g., after $l$ layers the representations of each node are dependent on their $l$-hop neighborhood, i.e., the nodes with a path length of up to $l$. While a single Chebyshev convolutional layer can be made to have an arbitrary receptive field by increasing the degree of the polynomials, multi-layered architectures have been observed to be

beneficial [48]. However by definition, the Chebyshev convolutional layer is not straightforward to implement with multi-dimensional edge features, but can utilise scalar edge weights [48].

**Temporal graph neural network layers.** After the graph convolutional layers, the graph snapshot at time $t$ is associated with the following hidden features:

$$H^{(t)} = G(\mathcal{H}_{gl}^{(t)}, \theta_l),$$

where $H_i^{(t)}$ are the features of node $i$ computed by the cascade of $l$ graph convolutional layers from the graph snapshot $\mathcal{G}^{(t)}$. A temporal neural network layer will integrate information regarding node $i$ from the previous timesteps to the current one, i.e., it can be represented as a function:

$$Z_i^{(t)} = f_{temp}(H_i^{(1)}, H_i^{(2)}, ..., H_i^{(t)}). \tag{3}$$

In this study, we consider two types of $f_{temp}$, which are the recurrent neural network and self-attention mechanisms.

The recurrent neural networks used in graph convolutional recurrent networks [43], variational graph recurrent neural networks [44], and ROLAND [46] can be represented by the classic recurrent neural network equation [49]:

$$Z_i^{(t)} = f(H_i^{(t)}, Z_i^{(t-1)}), \tag{4}$$

which defines $f_{temp}$ through recursive application of a neural network $f$ to the node state $H_i^{(t)}$ and the output of the previous iteration $Z_i^{(t-1)}$. Dynamic self-attention networks [45] utilise the masked self-attention mechanism, popularized by transformers in natural language processing [50], which is defined as:

$$Z_i^{(t)} = \sum_{\tau=1}^{t} a_\tau^{(t)} V_i^{(\tau)},$$

$$a_\tau^{(t)} = \frac{\exp\left((Q_i^{(t)})^\top K_i^{(\tau)}\right)}{\sum_{k=1}^{t} \exp\left((Q_i^{(t)})^\top K_i^{(k)}\right)},$$

$$Q_i^{(t)} = W_Q H_i^{(t)}, \quad K_i^{(t)} = W_K H_i^{(t)}, \quad V_i^{(t)} = W_V H_i^{(t)},$$

where $Q_i, K_i, V_i$ are the so-called queries, keys, and values, respectively, defined by linear transforms of the representations $H_i$ with the weights $W_Q, W_K, W_V$. The variant of self-attention presented here is called causally masked, as the output $Z_i^{(t)}$ is only dependent on the information up to timestep $t$.

**Edge attribute modeling.** Given a combination of graph convolutional and temporal layers, the output features $O^{(t)}$ of each node at a timestep $t$ will be used to predict the edge attributes of the next timestep $t + 1$. Specifically, as a last layer of a GNN, a readout layer $f_{out}$ is used to model the number of calls and SMS between two nodes. The predicted number of calls $\hat{c}$ and SMS $\hat{m}$ from node $s$ to node $d$ for timestep $t + 1$ are estimated with the layer as:

$$[\hat{c}_{s \to d}^{(t+1)}, \hat{m}_{s \to d}^{(t+1)}] = f_{out}(O_s^{(t)}, O_d^{(t)}). \tag{5}$$

There are two variants used in our set of GNNs. Namely, the inner-product decoder and a multilayer perceptron (MLP) readout layers. The inner-product decoder performs the following:

$$f_{out}(O_s^{(t)}, O_d^{(t)}) = [(S_c O_s^{(t)})^\top (D_c O_d^{(t)}), (S_m O_s^{(t)})^\top (D_m O_d^{(t)})], \qquad (6)$$

where $S$, $D$ are matrices that are used to project the output feature vectors to "source" and "destination" vectors in order to break the following symmetry $(O_i^{(t)})^\top (O_j^{(t)}) = (O_j^{(t)})^\top (O_i^{(t)})$. Without the symmetry breaking the number of calls and SMS from $s$ to $d$ would equal those from $d$ to $s$. There are two sets of $S$, $D$, i.e., $S_c, D_c$ for calls and $S_m, D_m$ for text messages, such that there are different representations for the prediction of calls and SMS. As for the MLP readout layer, it computes the following:

$$f_{out}(O_s^{(t)}, O_d^{(t)}) = MLP([O_s^{(t)}, O_d^{(t)}]),$$

i.e., it simply uses an MLP network that processes the concatenated representations of source and target nodes to output the predicted quantities of calls and SMS simultaneously.

In order to evaluate the performance of a GNN in the prediction of the quantities of calls and text messages, we require two types of edges. The first type of edges are the so-called positive edges, which are the observed edges within the data. The performance of the GNN for these edges tells us how well the model performs if given two nodes that we know to be connected. The second type of edges are called negative edges, which are synthetic, i.e., non-existent edges, that are used to evaluate if the model can correctly predict no mobile activity between nodes that are not connected. In the literature, the nodes for negative edges are typically randomly sampled, while making sure that we do not randomly select two nodes that are actually connected. However, in [42] it was argued that these "random negative edges" can be too easy to detect. They instead proposed alternative methods to evaluate negative edge performance, one of which is the so-called historical negative edges. The historical negative edges are edges that have a connection in the past, but not at the timestep of evaluation. In our results, we evaluate the GNN performance on positive, random negative, and historical negative edges.

**Graph neural network models and training.** We considered four recently proposed temporal GNNs that are suitable for snapshot -type temporal graphs, i.e., the graph convolutional recurrent network (GCRN) [43], variational graph recurrent neural network (VGRNN) [44], dynamic self-attention network (DySAT) [45], and ROLAND [46]. We performed extensive grid search for optimal hyperparameters of each model using the average error on the validation set as the performance measure. The hyperparameters we chose were the learning rate, loss weighting, and the number of neural network parameters. The optimal hyperparameters are reported in the paragraph that introduces each model. We additionally utilised early stopping with a patience of 20 epochs of non-improvement of validation performance.

During training, validation, and testing, we used a mini-batch of 100 subgraphs. The random negative edges were created in an online fashion, by randomly shuffling the neighbors of seed nodes of each subgraph, while ensuring that the sampled nodes were not actual neighbors of the seed nodes. In practice, we sampled 10 times as many random negative edges as there were positive edges for each seed node.

Before feeding the graph data to the GNNs, we applied data normalization methods. Specifically, the gender was encoded to binary, i.e., 0 or 1, whereas the age, latitude, and longitude were normalized to the range of [–1,1] based on the maximum and minimum values of the data. The edge attributes, i.e., the number of calls and SMS, were scaled to have a mean of zero and standard deviation of one based on the attributes in the training set. However, we maintained the original value range of the number of calls and SMS for the prediction targets.

**Graph convolutional recurrent networks** were proposed in [43]. It combines the Chebyshev type graph convolutional layers with a long short-term memory (LSTM) [51] recurrent model. Specifically, each matrix multiplication within

the LSTM is replaced with a Chebyshev convolutional layer, which can be thought to correspond to multiple parallel graph convolutions that are used as input to the recurrent model. As described in the previous section, the Chebyshev graph convolutional layer cannot utilise edge features. In order to integrate the edge feature information to the Chebyshev convolutions, we used an MLP neural network with two layers to map the four dimensional edge features to scalar edge weights, which were then used in the weighted adjacency matrix within the layer. The output of the MLP was constrained within the range of [0,1] using a sigmoid activation function. Similar to the seminal work, our GCRN model utilises a single graph convolutional LSTM unit with the Chebyshev coefficient $K = 3$. The dimensionality of all layers was set to 176 units. The output states of the LSTM were used with the inner-product decoder to predict the number of calls and SMS. The model was trained with mean squared error loss using the Adam optimizer [52] with a learning rate of 0.0003.

**The variational graph recurrent neural networks** were introduced in [44]. It combines the GCRN model with a variational autoencoder [53] for temporal graph adjacency matrix modeling. There are two key differences to GCRN. First is that instead of an LSTM, the temporal modeling is performed using a gated recurrent unit (GRU) [54], and second, that each node is associated with a stochastic Gaussian latent variable, which determines the formation of edges between the nodes. The model is trained using variational inference, where a prior and approximate posterior networks are used to enable efficient optimization of an evidence lower-bound. The approximate posterior has full information of the state of the network at timestep $t$, whereas the prior is a simple neural network dependent only on the GRU output at previous timestep [44]. The prior network output is finally used with the inner-product decoder to predict the number of calls and SMS of the timestep $t$. Similar to GCRN, the VGRNN is used with Chebyshev graph convolutional layers, and thus we used an MLP to map the edge features to edge weights. The hidden and latent dimensionalities were set to 176 units. The model was trained with the variational lower bound loss presented in the seminal work, however, using a multivariate normal, instead of a Bernoulli, distribution to model the counts of calls and SMS. We used the Adam optimizer [52] to calculate the parameter updates with a learning rate of 0.0003.

**Dynamic self-attention networks** were proposed in [45]. It uses the self-attention mechanism [50] for both graph convolutions and aggregation of temporal information. Specifically, DySAT uses so-called structural self-attention, which is a form of message-passing and aggregation, as a graph convolutional layer, and temporal masked self-attention to aggregate information from previous timesteps. Similar to GCRN and VGRNN, DySAT cannot utilise multidimensional edge features, and instead uses edge weights that we computed from edge features using an MLP network. The original implementation of DySAT included so-called positional encodings, which are learned vectors that enable temporal self-attention layer to recognize the order of the time-series. However, as our held-out data consists of graph snapshots after the last training timestep, the positional encodings that correspond to timesteps after December 31, 2008, would never be trained. As such, we replaced the positional encoding with ALiBi [55]. In short, it modifies the temporal self-attention, such that the attention weight between $t$ and $\tau$, i.e., $a_\tau^{(t)}$ is down-weighted in proportion to the difference $|t - \tau|$. As it is a relative positional encoding, it was found in the seminal work to enable transformers to generalize to sequences that are longer than in the training set. The final layer of the model is an inner product decoder that is used to predict the quantities of calls and text messages. As for the hyperparameters, we used a hidden dimensionality of 89, and trained the model with mean squared error loss using Adam optimizer [52] with a learning rate of 0.0003.

**ROLAND**, proposed in [46], incorporates multiple successful strategies from static, i.e., non-temporal, GNNs to its design. The seminal work found that utilising residual skip connections [56], batch normalization [57], and multiple recurrent layers resulted in improved performance. Additionally, and in contrast to the previously described models, ROLAND can also utilise multi-dimensional edge features through its unique message-passing and aggregation layer. The work studied various combinations and configurations of their model, but in this work we only consider the configuration that they found to have performed the best is most tasks. Specifically, the ROLAND model we use consists of a cascade of two ROLAND graph convolutional layers, followed by two blocks of alternating GRU and graph convolutional layers, and finally followed by two graph convolutional layers. The final module of the ROLAND model is an MLP readout layer, which is used to regress the number of calls and SMS. As for the hyperparameters of the ROLAND model, the hidden dimension

of each layer was set to 160 and the learning rate had a value of $3 \times 10^{-5}$. The model was trained with the mean squared error loss using Adam optimizer [52].

### Evaluation measures and procedures

In order to gain more insight into the properties of our data and to evaluate GNN performance in a comprehensive manner, we follow the methodology proposed in [42]. Specifically, the work proposed measures and data visualisation methods to reveal interesting properties about temporal graphs in relation to graph neural network -based analysis. The proposed measures were *novelty*, *reoccurrence*, and *surprise*, and the two data visualisation methods were *temporal edge appearance* (TEA) plot and *temporal edge traffic* (TET) plot. The measures and their purpose is as follows. Novelty is calculated as [42]:

$$novelty = \frac{1}{T} \sum_{t=1}^{T} \frac{|\mathcal{E}^{(t)} \backslash \mathcal{E}^{(<t)}|}{|\mathcal{E}^{(t)}|},$$

$$\mathcal{E}^{(<t)} = \{(s,d) \mid (s,d) \in \bigcup_{\tau=1}^{t-1} \mathcal{E}^{(\tau)}\},$$

where $\mathcal{E}^{(t)} \backslash \mathcal{E}^{(<t)}$ is the set of edges that are observed on timestep $t$, but not before. It thus measures the proportion of new edges appearing each month, averaged over the full time period. The TEA plot visualises novelty using a bar graph, where for each timestep the number of novel edges $|\mathcal{E}^{(t)} \backslash \mathcal{E}^{(<t)}|$ and the number of reoccurring edges $|\mathcal{E}^{(t)} \cap \mathcal{E}^{(<t)}|$ are shown. It thus illustrates how the novel and reoccurring edges are distributed in time.

When the temporal graph is divided to the development (i.e., training and validation) and test subsets based on a cutoff timestep $t_c$, the set of edges in the development set are $\mathcal{E}_{dev} = \bigcup_{\tau=1}^{t_c} \mathcal{E}^{(\tau)}$ and the set of edges in the test set are $\mathcal{E}_{test} = \bigcup_{\tau=t_c+1}^{T} \mathcal{E}^{(\tau)}$. The set of "transductive" edges, i.e., those that appear both during training and testing are $\mathcal{E}_{dev} \cap \mathcal{E}_{test}$, whereas the set of "inductive" edges are those that only appear in the test set $\mathcal{E}_{test} \backslash \mathcal{E}_{dev}$. The reoccurrence and surprise are defined as [42]:

$$reoccurrence = \frac{|\mathcal{E}_{dev} \cap \mathcal{E}_{test}|}{|\mathcal{E}_{dev}|},$$
$$surprise = \frac{|\mathcal{E}_{test} \backslash \mathcal{E}_{dev}|}{|\mathcal{E}_{test}|},$$

where the reoccurrence measures the proportion of edges that the GNN models have seen during training that reappear in the test set, whereas surprise measures the proportion of novel edges only seen in the test set. The TET plot partitions the edges into blocks based on the timestep of the first appearance of the edge, and within each block it sorts the edges based on the last timestep the edge is present in the data. It can be used for fast visual assessment of the proportions of train only, transductive, and inductive edges.

When the reoccurrence index is high and surprise is low, high performance can be achieved by simply memorizing the training data, i.e., without complex analysis of patterns within the data. To determine if temporal GNNs can outperform memorization of the data, in [42] a memorization -based baseline model called EdgeBank was proposed. The study only considers temporal edge prediction in the binary case, i.e., the prediction of the existence of an edge, whereas we consider the regression of the edge features, i.e., the number of calls and SMS. Thus, we propose a modified version of

EdgeBank for regression, which we denote as rEdgeBank. It predicts the edge features as:

$$\text{rEdgeBank}_w(i,j,t) = \frac{1}{|S_w^{(t-1)}|} \sum_{E_{i,j} \in S_w^{(t-1)}} E_{i,j},$$

$$S_w^{(t-1)} = \bigcup_{\tau=t-w}^{t-1} \{E_{i,j}^{(\tau)} \mid (i,j) \in \mathcal{E}^\tau\},$$

where $(i, j)$ denotes the edge between nodes $i$ and $j$, $t$ the timestep of the edge features to be predicted, $e_{i,j}$ the edge feature of edge $(i, j)$, and $w$ denotes the width of the time window that the model is allowed to remember the edges from. The rEdgeBank thus predicts the number of calls and SMS for each edge as the average observed in the previous $w$ timesteps. If there is no edge between $i$ and $j$ in the previous $w$ timesteps, we define the prediction as 0 for both calls and SMS. Similar to optimizing the hyperparameters of the GNNs, we performed grid search to find the optimal $w$, which we found to be four.

As the performance measure, i.e., the goodness of fit of the models, we use the mean absolute error (MAE) for both calls and SMS, defined as:

$$\text{MAE} = \frac{1}{\prod_{\tau=t_c+1}^{T} |\mathcal{E}^{(\tau)}|} \sum_{\tau=t_c+1}^{T} \sum_{(i,j) \in \mathcal{E}^{(\tau)}} |E_{i,j}^{(\tau)} - \hat{E}_{i,j}^{(\tau)}|,$$

where $t_c$ is the train cutoff timestamp, $E_{i,j}^{(\tau)}$ and $\hat{E}_{i,j}^{(\tau)}$ are the edge features and model predictions, respectively, at time $\tau$, the $|\cdot|$ is overloaded notation that denotes the cardinality of a set for set inputs and the absolute value when applied to scalars. MAE simply measures the error in terms of the mean absolute value of the difference between prediction and ground truth.

## Results

In this section, we present comprehensive analysis of our data and the results of the GNNs. First, we illustrate detailed analysis of our unique mobile phone dataset. Second, we present the results of the four temporal GNNs with comparisons to our proposed baseline, including detailed results when stratifying the performance based on age and gender.

### Dataset temporal edge analysis

The mobile phone dataset analysis reveals that vast majority of the edges persisted throughout the 36 month period considered in this study, as indicated by the low novelty index of 0.05. Furthermore, we obtained reoccurrence index of 0.78, which means that 78% of the edges seen in the first 30 month period, i.e., in the train and validation subsets, persisted to the final 6 month period, i.e., the test set. As for the surprise index, it turned out to have the value of 0.03 that shows very small number of test edges to be truly unique to the test set.

The results are illustrated with the TEA and TET plots, shown in Figs 2 and 3, respectively. In addition to visually presenting the low number of unique edges appearing in each month, i.e., the low novely index, TEA plot also reveals a seasonality pattern, where the winter month of December exhibits a large peak in the call and SMS activity. As for the TET plot, we can easily observe the high reoccurrence index through noting the large amount of edges with the color orange, which signify the edges that appear in both development and test sets. Furthermore, the test only, i.e., inductive, edges make up only a small portion of the total amount of edges that translates to the low surprise index. TET plot additionally reveals that approximately 50% of the edges appear already in the very first month of the dataset with a majority of them also seen in the final month.

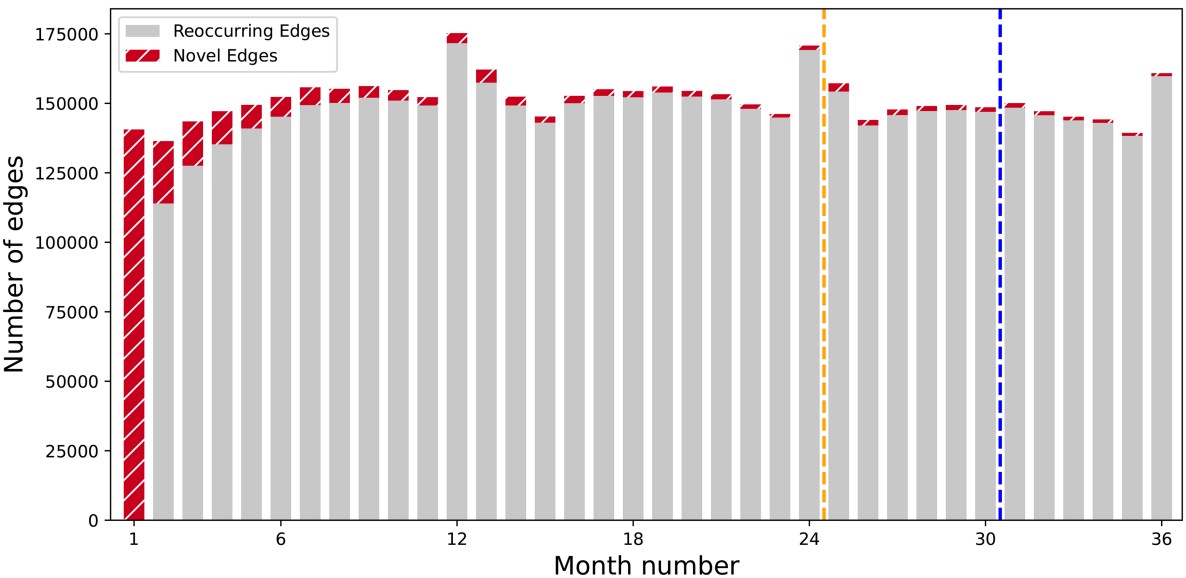

**Fig 2**. **Temporal Edge Appearance (TEA) plot.** For each month, the number of novel, i.e., yet unseen, edges are presented with the striped red bars, and the number of reoccurring, i.e., previously observed, edges are presented with gray bars. Novelty index is 0.05. Validation cutoff is denoted with an orange and test cutoff with a blue dashed line.

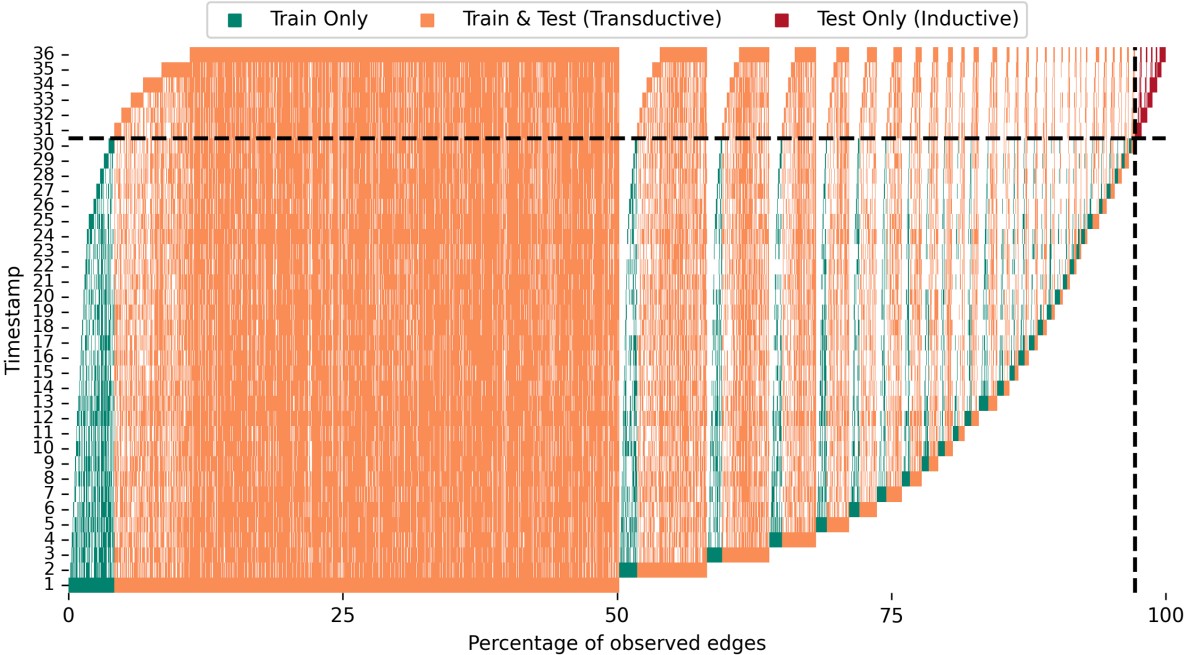

**Fig 3**. **Temporal Edge Traffic (TET) plot.** The edges of the dataset are visualized by sorting and grouping them based on the month of first appearance, and for each group, they are again sorted based on the last month of appearance. The edges are colored based on the group they belong to, i.e., green for training only edges, orange for edges in the training and test sets (transductive), and red for test only (inductive) edges. The reoccurrence index is 0.78 and surprise index is 0.03. Test cutoffs in time and percentage are denoted with black dashed lines.

## Graph edge regression

In this subsection, we first present the dataset level summary of the model performance for each GNN and the baseline, and then we turn to the performance stratified by age and gender.

**GNN model performance.** As for the prediction of the number of calls, it turned out that ROLAND had the lowest overall error, in terms of the MAE measure, of 1.09 and the lowest error for the positive edges with a value of 3.11 (4.98), while it was on par with the rEdgeBank for random negative edges with a mean of 0.00, although it had a larger standard deviation of 0.13. In the case of historical negative edges, GCRN had the lowest error with a value of 0.14 (1.00). When comparing to the rEdgeBank baseline, ROLAND was the only model to outperform it in terms of positive edge prediction, whereas in the case of historical negative edges, only DySAT had worse performance in comparison. As the rEdgeBank is based on memorization of the past data, it will always have zero error for random negative edges by definition.

When evaluating the performance in the prediction of the number of text messages, ROLAND outperformed the other GNNs and rEdgeBank in terms of the overall and positive edge error. Interestingly in this prediction task, every GNN obtained 0.00 average error for the random negative edges. As for the historical negative edge prediction performance, VGRNN turned out to have the best performance with a value of 0.00 (0.40). When comparing the models to the baseline, ROLAND was the only model to outperform rEdgeBank for the positive edges and in average performance. Every GNN also outperformed rEdgeBank for the historical negative edge prediction. The results are summarized in Table 1.

We also examined the model performance for the positive edges as a function of the timestep, i.e., the month, as illustrated in Fig 4. Interestingly, the errors are stable across the months, even though the number of edges spikes on December, as seen in Fig 2. We can see that ROLAND performs overall the best for each month with the exception of underperforming rEdgeBank baseline SMS performance for the month of December, i.e., month number 36.

**Result stratification on age and gender.** We stratified the positive edge, i.e., observed connection, results based on the characteristics of the source and destination nodes, according to the groups used in previous research [13]. The results for the gender stratification are presented in Table 2, and the age stratification results for calls in Fig 5 and for text messages in Fig 6. When the results were grouped based on gender, i.e., the prediction errors were grouped to female to female, female to male, male to female, and male to male types, ROLAND had consistently the best performance for every connection type and for both the calls and text messages. Furthermore, ROLAND was the only model that outperformed the rEdgeBank baseline. Interestingly, for rEdgeBank, GCRN, VGRNN, and ROLAND, the between gender edge prediction had lower error than the within gender prediction. This pattern can be seen to be reversed for text messages for the same set of models. We additionally performed temporal analysis for the gender stratified groups as presented in S1 Fig.

As for the age stratified call prediction results, ROLAND outperformed rEdgeBank for all age groups except for when source was of 45–55 and destination of 60–65. No other model could outperform rEdgeBank with the exception of GCRN

**Table 1.** **Mean Absolute Error (MAE) results grouped by the connection (Call and SMS) and evaluation types.** A lower MAE value indicates better performance, and the best results are shown in bold. MAE is decomposed to mean (STD) prediction error of positive edges (Positive), mean (STD) prediction error of random negative edges (R-Negative), mean (STD) prediction error of historical negative edges (H-Negative), and average of the three means (Ave). Using Wilcoxon signed-rank test revealed that all the GNN edge prediction results are statistically significantly different compared to the rEdgeBank baseline.

| | Mean Absolute Error | | | | | | | |
| | Call | | | | SMS | | | |
| Method | Positive | R-Negative | H-Negative | Ave | Positive | R-Negative | H-Negative | Ave |
| --- | --- | --- | --- | --- | --- | --- | --- | --- |
| rEdgeBank | 3.24 (5.30) | **0.00 (0.00)** | 0.24 (1.16) | 1.16 | 0.87 (3.98) | **0.00 (0.00)** | 0.07 (0.69) | 0.31 |
| GCRN | 3.75 (5.87) | 0.01 (0.20) | **0.14 (1.00)** | 1.30 | 1.14 (4.89) | **0.00 (0.09)** | 0.01 (0.39) | 0.38 |
| VGRNN | 5.55 (9.72) | 0.10 (0.79) | 0.16 (0.86) | 1.94 | 1.17 (5.78) | **0.00 (0.20)** | **0.00 (0.40)** | 0.39 |
| DySAT | 5.87 (10.84) | 0.15 (0.72) | 0.30 (1.22) | 2.11 | 1.23 (7.11) | **0.00 (0.15)** | 0.01 (0.35) | 0.41 |
| ROLAND | **3.11 (4.98)** | **0.00 (0.13)** | 0.15 (0.89) | **1.09** | **0.83 (3.68)** | **0.00 (0.04)** | 0.02 (0.42) | **0.28** |

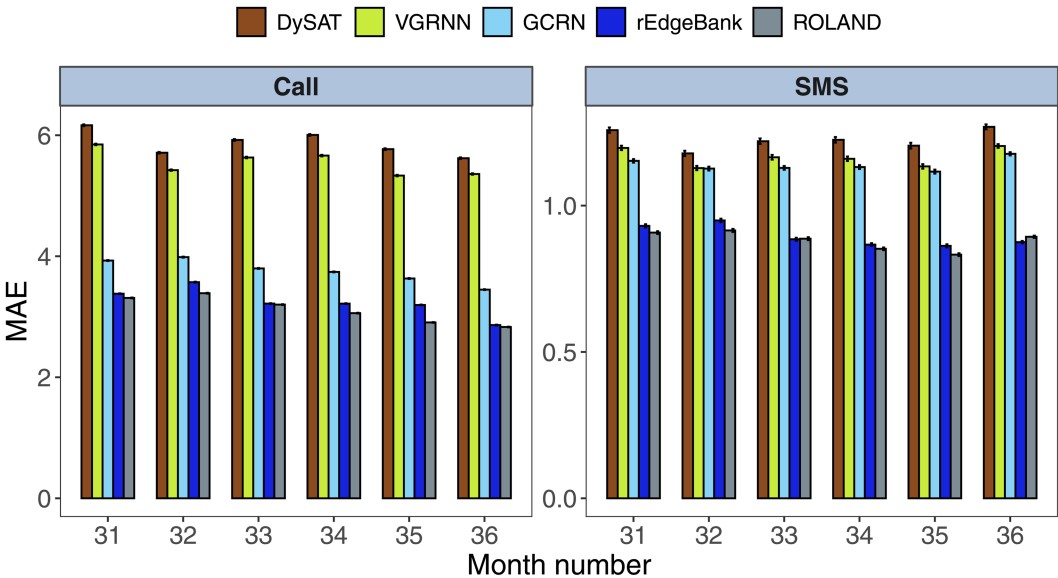

**Fig 4**. **The Mean Absolute Error (MAE) with respect to time.** The barplot illustrates the positive edge MAE values of each model with respect to the month in the test set.

**Table 2**. **Mean absolute error (MAE) results stratified by the connection type and gender.** The table displays the MAEs of call and SMS prediction performance, obtained for the positive edges, with respect to gender of the source and destination nodes.

| Connection | Method | Female to Female | Female to Male | Male to Female | Male to Male |
|---|---|---|---|---|---|
| Call | rEdgeBank | 3.32 (5.17) | 3.09 (5.54) | 3.09 (5.39) | 3.34 (5.21) |
| | GCRN | 3.79 (5.73) | 3.62 (6.09) | 3.62 (6.09) | 3.89 (5.72) |
| | VGRNN | 6.36 (11.06) | 4.26 (7.29) | 4.37 (7.46) | 6.31 (10.63) |
| | DySAT | 5.63 (9.97) | 6.00 (11.93) | 6.06 (11.91) | 5.97 (10.25) |
| | ROLAND | **3.19 (4.89)** | **2.97 (5.07)** | **2.98 (5.10)** | **3.20 (4.93)** |
| SMS | rEdgeBank | 0.67 (2.85) | 0.99 (4.97) | 1.07 (5.40) | 0.90 (3.11) |
| | GCRN | 0.87 (4.14) | 1.28 (5.72) | 1.43 (6.16) | 1.17 (4.01) |
| | VGRNN | 0.89 (5.55) | 1.29 (5.93) | 1.44 (6.37) | 1.22 (5.51) |
| | DySAT | 0.90 (5.44) | 1.44 (8.78) | 1.61 (9.53) | 1.21 (5.38) |
| | ROLAND | **0.64 (2.76)** | **0.93 (4.42)** | **1.01 (4.66)** | **0.87 (3.26)** |

for the edges where source was of 60–65 and destination of 18–21. The pattern of errors was remarkably similar for all models with the highest errors appearing within each age group and between young and old people.

When examining the age stratified text message prediction results, ROLAND outperformed rEdgeBank for most age groups with the exception of having the same performance when source was in the group 45–55 and destination in 60–65, and in the case when source was in the group 60–65 and target in 25–35. No other GNN outperformed rEdgebank for any of the groupings. It is worth noting that similarly as in the call prediction results, all the GNNs and the baseline exhibit similar error patterns, where the error increases as the source and destination nodes get younger.

## Concluding remarks

In this study, we have systematically analysed the temporal edge patterns of a large scale mobile phone dataset and implemented four temporal graph neural networks, i.e., GCRN, VGRNN, DySAT, and ROLAND, for the prediction of future call and text message activities. In addition, we have developed a simple but effective baseline method called rEdgeBank,

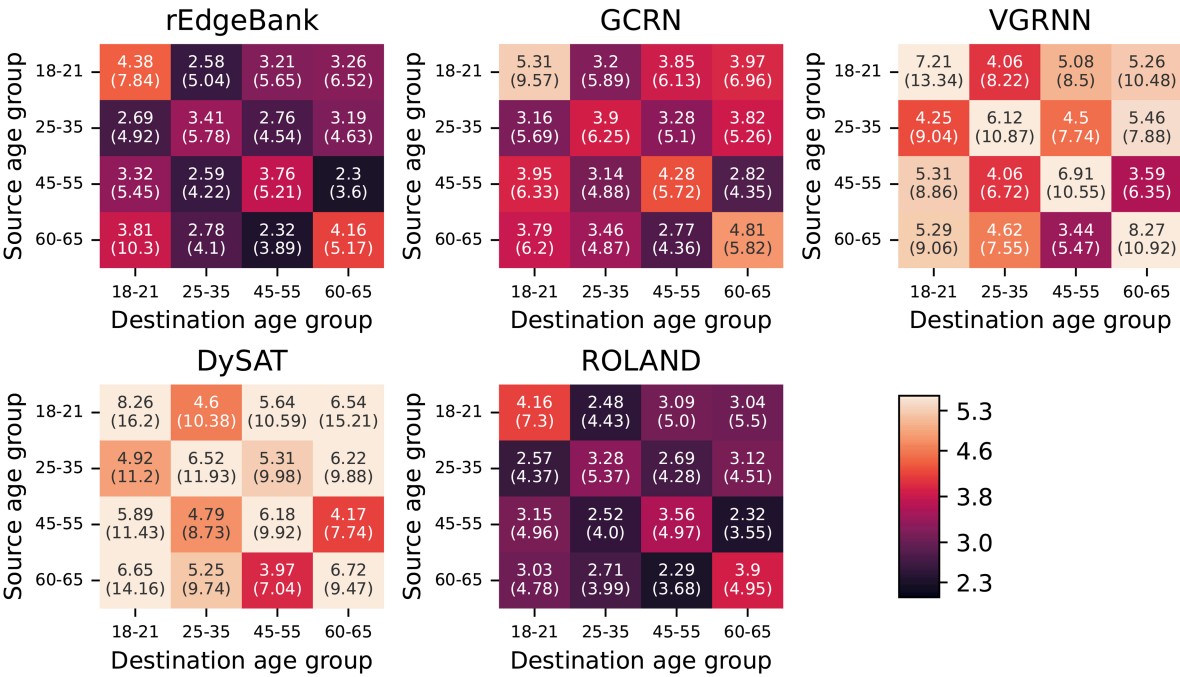

**Fig 5. MAE results for positive edge call prediction stratified based on the age of the source and destination nodes.** The MAE values for each possible pair of source and destination groups, i.e., age groups of 18–21, 25–35, 45–55, and 60–65, arranged in a grid. The cells are colored with the given colorbar, such that brighter color denotes larger error.

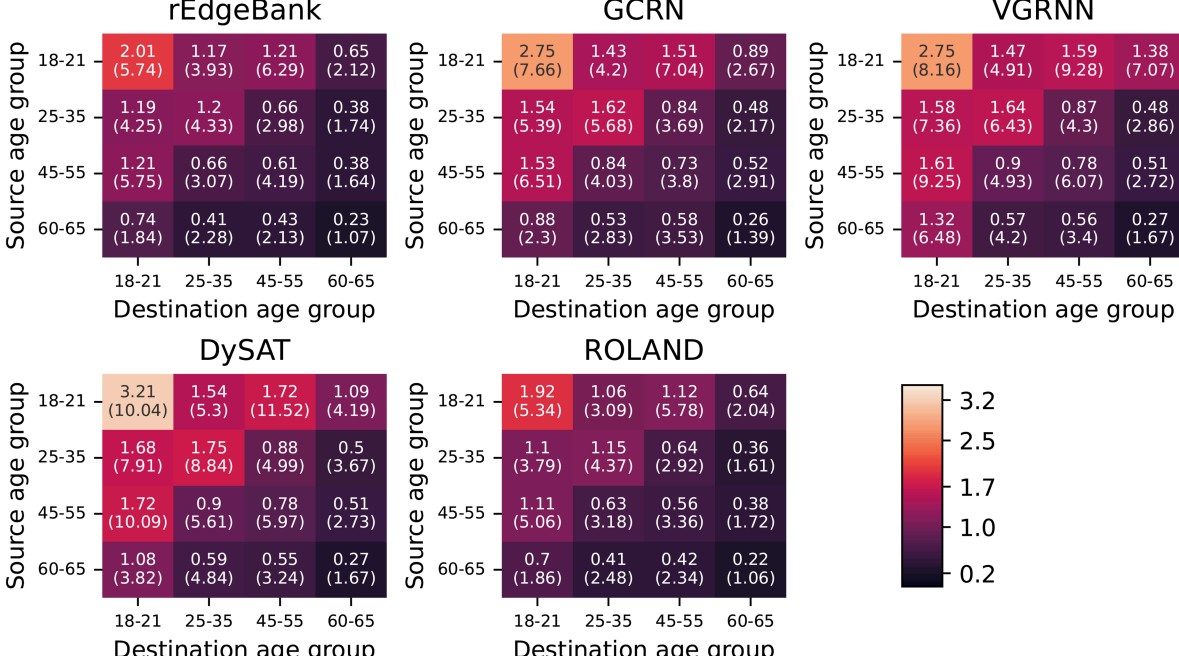

**Fig 6. MAE results for positive edge SMS prediction stratified based on the age of the source and destination nodes.** The MAE values for each possible pair of source and destination groups, i.e., age groups of 18–21, 25–35, 45–55, and 60–65, arranged in a grid. The cells are colored with the given colorbar, such that brighter color denotes larger error.

in order to evaluate the quality of the GNN predictions. Our results show that the ROLAND GNN had turned out to yield best results for the prediction task and it was the only GNN systematically outperforming the baseline.

In [42], it was observed that if a temporal graph dataset has both a high reoccurrence index, and a low novelty and surprise indices, a simple memorization - based approach can provide a strong baseline for edge prediction tasks. Our findings can be interpreted to yield a similar conclusion. Indeed, in comparison to the datasets considered in the original work, our mobile phone dataset had very low surprise and novelty indices, and a moderately high reoccurrence index. Furthermore, we found that rEdgeBank memorization -based baseline outperformed all GNNs, with the exception of ROLAND. It should be noted that the temporal resolution of our network data, i.e., the monthly graph snapshots, can also be a contributing factor to the efficacy of the rEdgeBank. Indeed, as the quantities of calls and text messages are aggregated on a monthly basis, they can be more stable than those measured on a weekly or daily basis. Increasing the time-resolution of the aggregated data could be investigated in the future to explore if the GNN -based approaches have advantage in this case.

When the edge prediction errors were considered, ROLAND mostly outperformed all the other GNNs and the baseline, but the difference between the performances of ROLAND and baseline were rather small. However, considering the expected high performance of the baseline for our dataset, the performance of ROLAND indicates that a temporal graph neural network approach can be used to learn predictive features that outperform simple memorization. Interestingly, the GNNs exhibited similar error patterns as rEdgeBank for the various stratification subsets. This could indicate that the prediction of mobile phone activity is inherently hard in some cases of social interactions, even when data driven approaches are used. In particular, the larger error values observed between younger caller-callee pairs when compared to their older counterparts could be due to the fact that young adults are better acquainted with technology [58] and therefore may tend to have a higher volume of calls and SMS. Furthermore, the turnover of personal networks in younger and middle-aged population is considerably higher than in older generations and therefore implies more fluctuations [12].

The efficacy of ROLAND could be due to its neural network architecture. Specifically, ROLAND is the only temporal GNN that can utilise vector -valued edge features, in our case the four features consisting of the bidirectional number of calls and text messages. The three other GNNs, i.e., GCRN, VGRNN, and DySAT, can only utilise scalar -valued edge weights, which we allowed these models to learn with a small multilayer perceptron network that takes the edge feature vector as an input. In the case of GCRN and VGRNN, this scalar weight appears in the graph Laplacian of the Chebyshev graph convolutional layer, whereas with DySAT it appears in the structural self-attention layer. In both cases, the scalar weight modulates the information flow between the nodes within the graph neural networks. This means that these scalar weights should simultaneously accomplish two tasks: 1) they should encode the four dimensional edge features to scalars such that this encoding is as invertible as possible to utilise the prior bidirectional call and SMS features in the prediction, and 2) when used to modulate the graph convolutions, they should allow information to flow between the nodes. This is in contrast to the graph convolutions of ROLAND, which do not compress these features or place restrictions on the information flow between the nodes based on the value of the edge features, but it can learn to do so if it is beneficial. Thus, ROLAND can more effectively utilise all the features of the network, i.e., both node and edge features. Indeed, the interpretation of the features of a link, i.e., the amount of calls and text messages, is influenced by the features of these individuals, e.g., their gender and age, and also their relationship that may be inferred from the past. For example, as seen in the case of non-peer relationships between women related to child rearing tasks among others [13,59]. Furthermore, the reciprocity of the mobile communication is apparent from the full link features. ROLAND is the only model that can distinguish between relationships when interpreting the edge features. While the models also differ by other architectural aspects than the definition of their graph convolutional layers, there are no other consistent architectural differences between the models to explain the performance gap with ROLAND. First, we performed extensive grid search over the optimal number of parameters for each GNN, thus the performance of each of the models is an estimate of the best performance they can obtain on the task. Second, when examining the GNN architectures in terms of the number of layers and their average performance, (i) GCRN with a single Chebyshev convolutional LSTM outperformed VGRNN and

DySAT, (ii) VGRNN with three layers outperformed DySAT with four layers, but (iii) ROLAND with 6 layers outperformed all the others. Thus, the amount of layers does not consistently improve the performance.

There are some limitations to our study. First, due to some people having different random unique identifiers for each year that could not be merged, we had to exclude people that did not have a consistent identifier for each year. This in turn had the effect that in addition to data associated with erroneous user identifiers, also all the data associated with users that had no activity for at least one full calendar year, i.e., from January to December, were excluded. This could have had the effect that the resulting network is more stable than if it had these users, which might be a contributing factor to the high performance of the baseline. However, as this filtering step only considered full calendar years, users could still have long periods of inactivity, e.g., from February 2007 to November 2008, without being filtered out that alleviates the issue. Second, the effects of comparably smaller networks or communities of the older populations on GNN training should be investigated to evaluate if there are biases in the results. Lastly, our set of data was gathered between 2007 and 2009 from a single country, which might limit the generalizability of our approach to recent data. However, as direct messaging applications were less used during that time, our dataset represents real-time interactions that reflect real social relationships. These relationships tend to remain quite stable throughout a person's life, especially with friends and family who form the core inner circle in Dunbar's layers of friendship, and this pattern has also been previously observed in our dataset [60]. We expect that if one had access to all forms of modern communication and combined the activity, the nature of friendships would still follow the social brain hypothesis [61] and thereby exhibit similar structural constraints including cognitive limitations for maintaining strong ties and a preference for having long term stable close relationships. As for the generalizability of the GNN -based approach for modern datasets, we believe that ROLAND would have an advantage in the prediction task. As the edge features of ROLAND can be simply expanded with more available link features, e.g., connectivity data from messaging or video call applications, it should have more capability than the other GNNs to predict future connectivity. In fact, the three other GNNs could perform worse on modern data as they require more link features to be compressed to the scalar representations. However, in the case of analysing only call and text messaging activity of today might be challenging due to the shift from traditional mobile communication to the messaging applications. This shift has caused especially the decline in the use of SMS that might make the prediction difficult due to its sparse usage.

To summarize, we have analysed the temporal edge patterns of a unique set of mobile phone data covering three years of call and text message activity. We systematically evaluated four state-of-the-art temporal graph neural networks for the prediction of future mobile phone activity. In addition, building on prior literature, we proposed a novel baseline approach to evaluate the models. Our results show that the ROLAND temporal graph neural network had generally favourable performance in comparison to other methods and the baseline. However, further study is required to develop novel methods that could outperform the baseline with a larger margin.

## Supporting information

**S1 Fig. Mean Absolute Error (MAE) with time for gender groups.** The MAEs calculated on the test data (last 6 months of the dataset) is shown with respect to gender of the source and the destination nodes.
(PDF)

**S2 Fig. Age and gender distribution of the users.** The distribution of the age of users for both males and females is shown. Users below the age of 18 and above 65 have not been considered for the present study.
(PDF)

## Author contributions

**Conceptualization:** Joel Jaskari, Chandreyee Roy, Fumiko Ogushi, Jaakko Sahlsten, Kimmo Kaski.

**Data curation:** Joel Jaskari, Chandreyee Roy, Fumiko Ogushi.

**Formal analysis:** Joel Jaskari, Chandreyee Roy, Fumiko Ogushi, Kimmo Kaski.

**Investigation:** Joel Jaskari, Chandreyee Roy, Kimmo Kaski.

**Methodology:** Joel Jaskari, Mikko Saukkoriipi, Jaakko Sahlsten.

**Project administration:** Kimmo Kaski.

**Resources:** Kimmo Kaski.

**Software:** Joel Jaskari.

**Supervision:** Fumiko Ogushi, Kimmo Kaski.

**Validation:** Joel Jaskari, Chandreyee Roy, Fumiko Ogushi, Mikko Saukkoriipi, Jaakko Sahlsten, Kimmo Kaski.

**Visualization:** Joel Jaskari, Chandreyee Roy.

**Writing – original draft:** Joel Jaskari, Chandreyee Roy, Fumiko Ogushi, Mikko Saukkoriipi, Jaakko Sahlsten, Kimmo Kaski.

**Writing – review & editing:** Joel Jaskari, Chandreyee Roy, Fumiko Ogushi, Mikko Saukkoriipi, Jaakko Sahlsten, Kimmo Kaski.

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
