## [Decision Letter · Decision Letter 0]

19 Sep 2025

PONE-D-25-45645Temporal social network modeling of mobile connectivity data with graph neural networksPLOS ONE

Dear Dr. Kaski,

Thank you for submitting your manuscript to PLOS ONE. After careful consideration, we feel that it has merit but does not fully meet PLOS ONE’s publication criteria as it currently stands. Therefore, we invite you to submit a revised version of the manuscript that addresses the points raised during the review process.

We look forward to receiving your revised manuscript.

Kind regards,

Guangyin Jin

Academic Editor

PLOS ONE

3. In the online submission form you indicate that your data is not available for proprietary reasons and have provided a contact point for accessing this data. Please note that your current contact point is a co-author on this manuscript. According to our Data Policy, the contact point must not be an author on the manuscript and must be an institutional contact, ideally not an individual. Please revise your data statement to a non-author institutional point of contact, such as a data access or ethics committee, and send this to us via return email. Please also include contact information for the third party organization, and please include the full citation of where the data can be found.

Reviewers' comments:

Reviewer's Responses to Questions

**Comments to the Author**

1. Is the manuscript technically sound, and do the data support the conclusions?

Reviewer #1: Yes

Reviewer #2: Partly

2. Has the statistical analysis been performed appropriately and rigorously?

Reviewer #1: Yes

Reviewer #2: Yes

3. Have the authors made all data underlying the findings in their manuscript fully available?

Reviewer #1: Yes

Reviewer #2: Yes

4. Is the manuscript presented in an intelligible fashion and written in standard English?

Reviewer #1: Yes

Reviewer #2: Yes

5. Review Comments to the Author

Reviewer #1: The authors compare and predict the phone call and SMS activity by several types of temporal GNN and a baseline rEdgeBank. The results show that ROLAND is the most effective for predicting their activity, comparing with other methods.

The motivation is understandable, and the results contribute to both academic and practical fields. On the other hand, as indicated, the prediction task itself may have become too simple. While rEdgeBank's high accuracy and low surprise are practical, it is possible that the task has become overly straightforward. In addition, the absence of detailed data disclosure due to NDAs and similar agreements is also weakness.

Overall, this is a study of quite high quality. As there is still a little potential for improvement, I recommend a minor revision.

Major comments:

1. At first reading, I mistakenly thought you would be proposing a new GNN architecture. For instance, explicitly stating in the introduction that you are comparing existing models, or revising section/subsection in Materials & Methods, would help reduce such misunderstandings.

2. I recommend adding a bit more context about the motivation for this prediction task. In what specific situations or practical challenges could SMS and call prediction be useful?

Minor commnets [if possible]:

3. I understand this is difficult due to the NDA, but even within the data used, knowing the distribution of gender and age would likely help advance our understanding.

4. I thought rEdgeBank's high accuracy was due to its coarse temporal resolution. Couldn't the GNN model achieve higher accuracy if the prediction task were performed on finer-grained temporal data?

Reviewer #2: This paper presents a comprehensive and timely investigation into the use of temporal graph neural networks (GNNs) for predicting communication activity in a large-scale, real-world mobile phone dataset. The study is well-structured, the methodology is thorough, and the problem it addresses is of significant interest to the network science and machine learning communities. The authors are to be commended for their rigorous evaluation of four distinct GNN architectures against a well-conceived baseline, and for providing a nuanced analysis stratified by user demographics. The work is a valuable contribution, and my comments below are intended to help further strengthen the manuscript.

1.A primary point that warrants further discussion is the potential impact of the data preprocessing steps on the study's main findings. The authors state that they filtered the dataset to include only users who were active in all three consecutive years of data collection. While the motivation to handle potential user ID changes is practical, this is a rather stringent criterion that likely removes a significant amount of natural user churn and network evolution. This filtering may have inadvertently curated a network core of exceptionally stable users. This could be a contributing factor to the observed temporal properties of the graph, namely the very high reoccurrence index of 0.78 and the low surprise index of 0.03. Consequently, the prediction task becomes more susceptible to success via memorization, which might explain the formidable performance of the rEdgeBank baseline. It would strengthen the manuscript if the authors could expand on this in their discussion, explicitly connecting this filtering choice to the dataset's temporal characteristics and considering how it might frame the relatively small performance margin achieved even by the best-performing GNN, ROLAND.

2.Another intriguing aspect of the results is the disparity in performance among the GNNs themselves. The finding that only ROLAND managed to consistently outperform the rEdgeBank baseline is significant. The authors' hypothesis that ROLAND's success stems from its native ability to utilize multi-dimensional edge features is compelling and well-argued. This point could be elaborated upon further. For example, how might the model leverage the distinct bidirectional call and SMS features to capture richer social dynamics, such as the patterns of reciprocity mentioned in the introduction? Conversely, the underperformance of the other three state-of-the-art GNNs is a noteworthy result. A deeper reflection on why these models failed would be beneficial. Was the information loss from compressing edge features into a single scalar weight the critical handicap, or are there other architectural limitations that make them less suitable for this type of social interaction data? A more detailed discussion of these model-specific successes and failures would provide valuable insights for future research in this area.

3.Finally, the authors rightly acknowledge the age of the dataset (2007-2009) as a limitation. Their perspective that this dataset provides a unique window into communication patterns before the widespread adoption of over-the-top messaging applications is valid and interesting. To enhance the paper's forward-looking relevance, it would be helpful to include a brief, speculative discussion on the generalizability of these findings to contemporary communication networks. Given the dramatic shifts in how people communicate (e.g., the decline of SMS, the rise of encrypted messaging apps), how might the feature representations and model architectures need to evolve? For instance, would the clear distinction between calls and texts, seemingly vital for ROLAND's performance, retain its predictive power in a modern context? Adding some thoughts on these future challenges would round out the discussion nicely.

6. PLOS authors have the option to publish the peer review history of their article (what does this mean?). If published, this will include your full peer review and any attached files.

Reviewer #1: **Yes: **Masaki Chujyo

Reviewer #2: No

---

## [Author Response · Author response to Decision Letter 1]

6 Oct 2025

Response to Reviewers’ comments

We thank the Reviewers for thorough reading of our manuscript and constructive feedback for improving the manuscript. Changes made to the revised manuscript based on the comments by Reviewer #1 and Reviewer #2 are marked in yellow and green, respectively, as well the corresponding line rows of the revised text is noted. Our responses to point-by-point remarks below are in blue.

Reviewer #1:

The authors compare and predict the phone call and SMS activity by several types of temporal GNN and a baseline rEdgeBank. The results show that ROLAND is the most effective for predicting their activity, comparing with other methods.

The motivation is understandable, and the results contribute to both academic and practical fields. On the other hand, as indicated, the prediction task itself may have become too simple. While rEdgeBank's high accuracy and low surprise are practical, it is possible that the task has become overly straightforward. In addition, the absence of detailed data disclosure due to NDAs and similar agreements is also weakness.

Overall, this is a study of quite high quality. As there is still a little potential for improvement, I recommend a minor revision.

Our response: We want to thank the reviewer for expressing interest in our manuscript, and for the thorough analysis and comments of our manuscript.

Major comments:

1. At first reading, I mistakenly thought you would be proposing a new GNN architecture. For instance, explicitly stating in the introduction that you are comparing existing models, or revising section/subsection in Materials & Methods, would help reduce such misunderstandings.

Our response: We thank the reviewer for pointing out this ambiguity. We have now clarified that the GNNs are based on prior literature in the Abstract, Introduction, and Materials and methods.

The line numbers of the changes: 63-64, 226-229, and abstract lines 5-6

2. I recommend adding a bit more context about the motivation for this prediction task. In what specific situations or practical challenges could SMS and call prediction be useful?

Our response: Modeling and predicting the temporal patterns of a social network of mobile activity is of interest in various ways. There could be direct benefits of such approaches in e.g., quantifying the mobile network traffic, and social and societal recommendations, but perhaps more interestingly, a data-driven method could be utilised to advance our understanding of the function of these networks. Indeed, while various dynamics of these networks may be observed through examining the datasets directly, GNNs could be probed to see the effects of e.g., previous interactions and individuals’ age/gender to future mobile activity. However, the present study highlighted the need for more accurate temporal graph neural networks for the mobile network activity prediction task in order to have more confidence in such downstream analyses, which is left for future work.

We have amended the introduction to include these aspects.

The line numbers of the changes: 50-53

Minor comments:

3. I understand this is difficult due to the NDA, but even within the data used, knowing the distribution of gender and age would likely help advance our understanding.

Our response: We agree with the reviewer. We have added the age and gender distribution of the subscribers as a supplementary figure (Fig. S2_fig.eps)

4. I thought rEdgeBank's high accuracy was due to its coarse temporal resolution. Couldn't the GNN model achieve higher accuracy if the prediction task were performed on finer-grained temporal data?

Our response: We appreciate this proposition, which after consideration we agree as one possible cause for the high performance of rEdgeBank. The temporal graph indices that have been recently proposed highlight the stability of the network with respect to the links, i.e., most of the links in the test portion of the data were already formed in the training portion, which explains the good performance of memorization -based methods. However, it is also true that the monthly aggregated data removes some variability, i.e., the number of calls and SMS might be stable on a month-by-month basis, but when examined on a daily basis they might distribute to the days of the months. Thus on one hand, the coarse time-scale can favor memorization, but on the other hand, the GNN models can utilise the network to the prediction task, which in principle should allow them to be at least as good as examining only the past information about the links that rEdgeBank does.

We have expanded our discussion of the high performance of rEdgeBank based on this observation, and also remarked on increasing the time-resolution of the network snapshots as an avenue of future investigation.

The line numbers of the changes: 464-469

Reviewer #2:

This paper presents a comprehensive and timely investigation into the use of temporal graph neural networks (GNNs) for predicting communication activity in a large-scale, real-world mobile phone dataset. The study is well-structured, the methodology is thorough, and the problem it addresses is of significant interest to the network science and machine learning communities. The authors are to be commended for their rigorous evaluation of four distinct GNN architectures against a well-conceived baseline, and for providing a nuanced analysis stratified by user demographics. The work is a valuable contribution, and my comments below are intended to help further strengthen the manuscript.

Our response: We thank the reviewer for their encouraging comments and suggestions made to improve it.

1. A primary point that warrants further discussion is the potential impact of the data preprocessing steps on the study's main findings. The authors state that they filtered the dataset to include only users who were active in all three consecutive years of data collection. While the motivation to handle potential user ID changes is practical, this is a rather stringent criterion that likely removes a significant amount of natural user churn and network evolution. This filtering may have inadvertently curated a network core of exceptionally stable users. This could be a contributing factor to the observed temporal properties of the graph, namely the very high reoccurrence index of 0.78 and the low surprise index of 0.03. Consequently, the prediction task becomes more susceptible to success via memorization, which might explain the formidable performance of the rEdgeBank baseline. It would strengthen the manuscript if the authors could expand on this in their discussion, explicitly connecting this filtering choice to the dataset's temporal characteristics and considering how it might frame the relatively small performance margin achieved even by the best-performing GNN, ROLAND.

Our response:

We appreciate the Reviewer raising these concerns regarding the effects of the user ID filtering preprocessing step. As the Reviewer mentioned, the filtering step is likely contributing to the temporal graph indices due to it leaving out users that enter the service in the final 6 months of the data, i.e., the test portion, which directly affects the reoccurrence and surprise indices. As regarding the possibility of the filtering step to have curated the network to a stable core, it is also possible, but we cannot verify it. A minor detail that might alleviate the amount of “false positive” IDs filtered out was that we performed the filtering based on calendar years of inactivity as opposed to on a monthly basis. Thus, there can be users that are very seldom active, even for over a year, as long as the period of inactivity is not a full calendar year but e.g., from Feb 2007 to Nov 2008.

We have discussed the effects of the filtering more in detail in the Concluding remarks section.

The line numbers of the changes: 522-529

2. Another intriguing aspect of the results is the disparity in performance among the GNNs themselves. The finding that only ROLAND managed to consistently outperform the rEdgeBank baseline is significant. The authors' hypothesis that ROLAND's success stems from its native ability to utilize multi-dimensional edge features is compelling and well-argued. This point could be elaborated upon further. For example, how might the model leverage the distinct bidirectional call and SMS features to capture richer social dynamics, such as the patterns of reciprocity mentioned in the introduction? Conversely, the underperformance of the other three state-of-the-art GNNs is a noteworthy result. A deeper reflection on why these models failed would be beneficial. Was the information loss from compressing edge features into a single scalar weight the critical handicap, or are there other architectural limitations that make them less suitable for this type of social interaction data? A more detailed discussion of these model-specific successes and failures would provide valuable insights for future research in this area.

Our response: The remarks and suggestions by the Reviewer are valuable for the purpose of our manuscript, and we agree that addressing them would further strengthen the interpretation of our results.

We first address the differences in the modeling capability of multi-dimensional edge features, used by ROLAND, and scalar ones, used by GCRN, VGRNN, and DySAT. As we described in the manuscript, the scalar edge weights were learned using a standard feed-forward neural network as a layer within each GNN. In the case of GCRN and VGRNN, this scalar weight appears in the graph Laplacian of the Chebyshev graph convolutional layer, whereas with DySAT it appears in the structural self-attention layer. In both cases, the scalar weight modulates the information flow between the nodes within the graph neural networks. This means that these scalar weights should simultaneously accomplish two tasks: 1) they should encode the four dimensional edge features to scalars such that this encoding is as invertible as possible to utilise the prior bidirectional call and SMS features in the prediction, and 2) when used to modulate the graph convolutions, they should allow information to flow between the nodes. This is in contrast to the graph convolutions of ROLAND, which places no restrictions on information flow between the nodes based on the value of the edge features (it can learn to do it if it is beneficial) and it does not compress these features. Thus, ROLAND can more effectively utilise all the features of the network without compromising representational quality. As regarding the reciprocity of the mobile activity, the same principles can have an effect.

As for other architectural aspects that might limit the performance of other GNNs, we performed a hyperparameter search over the number of parameters of each model that was carefully set so that each model had the opportunity to have the same number of parameters. The optimal number was selected based on validation set performance. This procedure is detailed in the manuscript “Graph neural network models and training” subsection. Thus the number of parameters is likely not the limiting factor. Another limiting factor could be the depth of the models, i.e., how many graph convolutional and temporal layers they have. However, there is no consistent pattern regarding this aspect, as while ROLAND does have the most layers with a total of 6, DySAT has 4, VGRNN has 3, and GCRN has 1, but GCRN has still more favourable performance than DySAT and VGRNN on average. It is also difficult to compare the number of layers as all the models have slightly different definitions of the layers and a single Chebyshev convolutional layer is more expressive than the message passing layers of DySAT and ROLAND (without considering the edge features).

Thus, in conclusion, we believe that the major cause of the difference in performance of ROLAND and the other GNNs is how they use the edge features.

We have included these aspects in the discussion.

The line numbers of the changes: 490-503, 507-508, 509-519

3. Finally, the authors rightly acknowledge the age of the dataset (2007-2009) as a limitation. Their perspective that this dataset provides a unique window into communication patterns before the widespread adoption of over-the-top messaging applications is valid and interesting. To enhance the paper's forward-looking relevance, it would be helpful to include a brief, speculative discussion on the generalizability of these findings to contemporary communication networks. Given the dramatic shifts in how people communicate (e.g., the decline of SMS, the rise of encrypted messaging apps), how might the feature representations and model architectures need to evolve? For instance, would the clear distinction between calls and texts, seemingly vital for ROLAND's performance, retain its predictive power in a modern context? Adding some thoughts on these future challenges would round out the discussion nicely.

Our response: We thank the reviewer for this comment. Indeed, modern communication includes several different types of channels, however, the relationships between individuals tend to remain quite stable throughout a person’s life, especially with friends and family irrespective of the communication methods. The social brain hypothesis proposed by Robin Dunbar suggests that the nature of friendships follow a layered structure and this pattern has also been previously observed in our dataset [60]. We expect that if one had access to all forms of modern communication and combined the activity, the core friendships [61] would exhibit similar structural constraints that include cognitive limitations for maintaining strong ties, and a preference for having long term close stable relationships.

Additionally, with access to datasets that combine both mobile and application -based communication, ROLAND should still be a generalizable approach, as the bidirectional edge features could be expanded to more than 4 dimensions to include the bidirectional features of other communication channels. Furthermore, a GNN -based approach could have further gains in these datasets due to multi-layeredness of the social networks. For example, a person might use a certain messaging application with friends and another with family, but might still use traditional mobile calls with both, which can be modeled more effectively with ROLAND in comparison to the baseline. However, if only modern mobile communication data is used, it is true that the distinction with calls and SMS might not make much sense due to the low number of text messages sent. In addition, the text message prediction task might be difficult for GNNs to learn due to the sparsity of today’s SMS data.

We have amended the limitations paragraph to also include this discussion.

The line numbers of the changes: 535-552

References

[60] Mac Carron P, Kaski K, Dunbar R. Calling Dunbar’s numbers. Social Networks.

2016;47:151–155.

[61] Dunbar R. Friends: Understanding the power of our most important

relationships. Hachette UK; 2021.

---

## [Decision Letter · Decision Letter 1]

8 Oct 2025

Temporal social network modeling of mobile connectivity data with graph neural networks

PONE-D-25-45645R1

Dear Dr. Kaski,

We’re pleased to inform you that your manuscript has been judged scientifically suitable for publication and will be formally accepted for publication once it meets all outstanding technical requirements.

Kind regards,

Guangyin Jin

Academic Editor

PLOS ONE

Additional Editor Comments (optional):

Reviewers' comments:

Reviewer's Responses to Questions

**Comments to the Author**

1. If the authors have adequately addressed your comments raised in a previous round of review and you feel that this manuscript is now acceptable for publication, you may indicate that here to bypass the “Comments to the Author” section, enter your conflict of interest statement in the “Confidential to Editor” section, and submit your "Accept" recommendation.

Reviewer #1: All comments have been addressed

2. Is the manuscript technically sound, and do the data support the conclusions?

Reviewer #1: Yes

3. Has the statistical analysis been performed appropriately and rigorously?

Reviewer #1: Yes

4. Have the authors made all data underlying the findings in their manuscript fully available?

Reviewer #1: Yes

5. Is the manuscript presented in an intelligible fashion and written in standard English?

Reviewer #1: Yes

6. Review Comments to the Author

Reviewer #1: The manuscript has been strengthened by the author's responses to the reviewers' comments, so I recommend accepting this manuscript for publication in PLOS ONE.

7. PLOS authors have the option to publish the peer review history of their article (what does this mean?). If published, this will include your full peer review and any attached files.

Reviewer #1: **Yes: **Masaki Chujyo

---

## [Editor Report · Acceptance letter]

PONE-D-25-45645R1

PLOS ONE

Dear Dr. Kaski,

I'm pleased to inform you that your manuscript has been deemed suitable for publication in PLOS ONE. Congratulations! Your manuscript is now being handed over to our production team.

Kind regards,

on behalf of

Dr. Guangyin Jin

Academic Editor

PLOS ONE